# Role of SAMITAL in the Prevention and Treatment of Chemo-Radiotherapy-Induced Oral Mucositis in Head and Neck Carcinoma: A Phase 2, Randomized, Double-Blind, Placebo-Controlled Clinical Trial (ROSAM)

**DOI:** 10.3390/cancers14246192

**Published:** 2022-12-15

**Authors:** Elena Fasanaro, Paola Del Bianco, Elena Groff, Antonella Riva, Giovanna Petrangolini, Fabio Busato, Paola Stritoni, Giovanni Scarzello, Lucio Loreggian, Gian Luca De Salvo

**Affiliations:** 1Otolaryngology Unit, Department of Surgery, Ospedale S. Antonio, Azienda Ospedaliera–Universitaria, 35127 Padova, Italy; 2Clinical Research Unit, Istituto Oncologico Veneto IOV–IRCCS, 35128 Padova, Italy; 3Radiotherapy Unit, Istituto Oncologico Veneto IOV–IRCCS, 35128 Padova, Italy; 4Product Innovation and Development Department, Indena SpA, 20139 Milano, Italy; 5Otolaryngology Unit, Ospedale dell’Angelo, 30174 Venezia Mestre, Italy

**Keywords:** head and neck squamous cell carcinoma, oral mucositis, SAMITAL, chemo-radiotherapy, botanical drug, quality of life

## Abstract

**Simple Summary:**

This Phase 2 double-blind placebo-controlled clinical trial was conducted in 116 stage III/IV head-neck squamous cell carcinoma patients to evaluate the role of SAMITAL in reducing the incidence of severe mucositis induced by concurrent chemo-radiotherapy. SAMITAL does not significantly reduce the incidence of severe mucositis; However, the lower rate of mucositis, together with a significantly better quality of life, suggested that a clinical benefit existed, and further research is needed to validate these findings. This trial is registered with the EU Clinical Trials Register database, number 2012-002046-20, and with ClinicalTrials.gov, NCT01941992.

**Abstract:**

Background: In patients affected by head and neck squamous cell carcinoma, the onset of severe oral mucositis is a decisive factor in completing concurrent chemo-radiotherapy, and few interventions have demonstrated a modest benefit. The primary aim of this clinical study was to evaluate the role of SAMITAL in reducing the incidence of severe mucositis induced by concurrent chemo-radiotherapy; the secondary aims were the tolerability and patient-reported quality of life measures. Methods: Patients were randomized to receive SAMITAL granules for oral suspension of 20 mL, four-time daily or matching placebo in a 1:1 fashion using a stratified-block randomization scheme by disease site and type of chemotherapy. The SAMITAL/placebo was dispensed at the baseline visit and at each weekly visit following radiotherapy initiation. Patients were subjected to weekly endoscopic evaluations to assess the presence of mucositis. In addition, patient-reported outcomes were measured. Results: Among the 116 patients treated with a median total dose of 66 Gy, 59 were randomized to SAMITAL and 57 to placebo. Overall, the incidence of severe mucositis was 51.7%, with 45.8% in the SAMITAL and 57.9% in the placebo arm (OR = 0.6; 95% CI: 0.3–1.3). After chemo-radiotherapy, patients randomized to SAMITAL reported significantly lower xerostomia, coughing and swallowing scores and a better quality of life. Conclusion: SAMITAL did not significantly reduce the incidence of severe mucositis in all studied populations. However, the lower rate of mucositis, together with a significantly better quality of life, suggested that a clinical benefit existed. This trial is registered with the EU Clinical Trials Register database, number 2012-002046-20, and with ClinicalTrials.gov, NCT01941992.

## 1. Introduction

The global burden of cancer continues to increase largely because of the aging and growth of the world population, alongside an increasing adoption of cancer-causing behavior (particularly smoking) in economically developing countries [1]. In 2021, Ferlay and colleagues reported the estimated worldwide cancer incidence for 2018, which was prepared by the International Agency for Research on Cancer (IARC) for 185 countries on five continents as part of GLOBOCAN 2018; an estimated 650,000 new cases ranked as the eight most common cancers worldwide, including head and neck carcinoma squamous cell carcinoma (HNSCC) (involving the lip, oral cavity, nasopharynx, other pharyngeal sites and larynx) [2]. The standard treatment for III–IV M0-staged resectable HNSCC is surgery with or without adjuvant radiotherapy plus concomitant chemotherapy [3]. Postoperative (chemo)-radiation improves tumor control and survival in HNSCC high-risk patients based on established risk factors [4], but it can also cause significant collateral effects. Chemo-radiotherapy treatment for HNSCC is characterized by early and delayed collateral effects; oral mucositis, xerostomia and dysphagia are among the least tolerated by patients. Oral mucositis is an inflammatory and/or ulcerative lesion of oral mucosa that affects 30–60% of patients receiving radiotherapy for HNSCC [5], causing pain, dysphagia often requiring feeding tube placement, treatment interruption, hospitalization and worsening of the patient’s quality of life (QoL).

Although current treatment for mucositis in HNSCC includes different therapeutic approaches [6,7], clinical management is often suboptimal. SAMITAL is a botanical drug composed of three highly standardized and purified botanical extracts formulated in sachets to be dispersed in water in a gel-like suspension to treat mucositis [8]. SAMITAL has been developed as a rational combination and is shown to be endowed with antibacterial, antifungal, antiviral, healing and anti-inflammatory activities, as previously reported [9,10,11,12].

This randomized, double-blind clinical trial was designed to evaluate the role of SAMITAL in reducing the incidence of severe chemo-radiotherapy-induced mucositis, as measured by the World Health Organization (WHO) scale. Our secondary aims were to evaluate the safety and tolerability of SAMITAL and its effect on reported symptoms in terms of severity and duration of mucositis, xerostomia and QoL measures according to the European Organization for Research and Treatment of Cancer: EORTC QLQ-C30 [13,14] and EORTC QLQ-H&N35 [15,16]. This trial is registered with the EU Clinical Trials Register database, number 2012-002046-20, and with ClinicalTrials.gov, NCT01941992.

## 2. Materials and Methods

### 2.1. Study Design and Participants

The study was designed as a prospective, single-center, double-blind, randomized and placebo-controlled Phase 2 clinical trial. Eligible patients were adults aged 18 years or older with: a histologically proven, stage III/IV HNSCC treated with definitive concurrent chemotherapy (induction or concomitant) and radiotherapy as intensity-modulated radiation therapy (IMRT) or conventional three-dimensional radiation therapy (C3DRT), Karnofsky performance status (KPS) ≥ 70, life expectancy ≥ 6 months, ability to take oral medication and swallow, availability to attend follow-up visits and willingness to use acceptable contraceptive methods during treatment (e.g., double barrier) for patients of child-bearing potential. Exclusion criteria were: previous radiotherapy involving oral and/or oropharyngeal mucosa, unwillingness or inability to follow protocol requirements, use of chronic immunosuppressive drugs or steroid therapy, pregnancy or nursing for female patients, presence of significant concomitant medical conditions (e.g., uncontrolled cardiac disorders, myocardial infarction within the 6 months prior to attending the study, history of severe neurological or psychiatric disorders) or ill-fitting dental appliances.

Patients were advised about daily oral hygiene and abstinence from tobacco and alcohol during the study. Odontoiatric evaluation was required at study entry to evaluate the necessity for root canal treatment or tooth extraction. For dental and cheek care, the use of a soft toothbrush was recommended. Mouthwash, but without alcohol and Vitamin E, and rinsing the mouth with water and baking soda before and after every meal were suggested

### 2.2. Randomization and Masking

Eligible patients were randomized in a 1:1 ratio to SAMITAL (granules for oral suspension of 20 mL, four times daily) or matching placebo. The placebo for SAMITAL granules for suspension and its reconstituted form were almost the same colors as the drug product and its reconstituted form, respectively. Flavor matching between SAMITAL and matching placebo was obtained by using the same sweetening agents in the two compositions. The placebo was packaged in the same container and prepared for administration in the same manner as the SAMITAL.

The randomization was carried out by the IOV Clinical Research Unit, using a permuted block scheme with a fixed block size of 4, stratified by disease site (oropharynx and oral cavity versus other) and type of chemotherapy (induction versus concomitant). Treatment allocation was blinded to patients, clinical investigators, data managers and study statisticians until the final study results.

### 2.3. Experimental Treatments

SAMITAL granules for suspension and the matching placebo were produced in Germany by Temmler under the current Good Manufacturing Practice (cGMP). The three standardized extracts constituent of the botanic drug used in the clinical trial were manufactured in Italy by Indena SpA under cGMP. SAMITAL was presented as single-dose paper/polyethylene (PE)/aluminum/PE sachets, each containing 1.5 g total weight. Of this, the extract preparations constituted 43.3 mg (Vaccinium myrtillus 40 mg, Macleaya cordata 2.7 mg and Echinacea angustifolia 0.6 mg). The inactive ingredients included a viscosity-increasing agent (guar gum), sweetening agents (sodium cyclamate, ammonium glycyrrhizinate and acesulfame potassium), an acidifier (anhydrous citric acid), a binding agent (hydroxypropyl cellulose), a surfactant (Tween 80) and diluents (mannitols). The placebo contained the same inactive ingredients and two food-grade coloring agents, FD&C Red No.40 (Allura Red, E129) and FD&C Blue No 1 (E133), for color matching.

SAMITAL/placebo was provided to patients at the first clinical evaluation at the beginning of radiotherapy and at each weekly visit until the end of radiotherapy when patients were provided SAMITAL/placebo for 4 more weeks (11 weeks overall since the beginning of radiotherapy). Patients were instructed to use SAMITAL/placebo four times daily, as in previous studies [8,10,11,12]. Briefly, the content of one sachet was mixed with 20 mL of water at room temperature and stirred until the powder was fully dispersed. The suspension was left to stand for at least 15 min until thickened. Patients were instructed to keep the liquid in different aliquots in their mouth and swish it around for at least 1 min, then either spit it out or swallow and not to drink or rinse their mouth for at least 10 min after SAMITAL/placebo administration.

Treatment was continued until discontinuation for one of the following reasons: death, inability to tolerate the oral study agent, toxicity unrelated to treatment, investigators’ judgment or patients’ voluntary withdrawal.

All patients received radiation at the same institution with concurrent chemotherapy as follows: For patients treated without surgery, the standard of care consisted of 70 Gy to the gross tumor in 35 fractions; regions at risk received a minimum of 60 Gy in 35 fractions (C3DRT or IMRT). For patients treated post-operatively, the standard of care consisted of 60–64 Gy to the area of the previous gross tumor in 30–32 fractions; regions at risk received a minimum of 50–54 Gy in 25–28 fractions. Chemotherapy consisted of platinum salts with or without 5-fluorouracil.

### 2.4. Assessments

Patients were assessed for mucositis by the otolaryngologist through oral exploration and video-endoscopy of the upper airways at baseline; assessments were conducted weekly during treatment (weeks 0–7) and at 4 weeks (week 11) and 3 months (week 19) after the end of radiotherapy. The incidence of severe mucositis was measured using the WHO score, the Oral Assessment Guide (OAG) [17,18] and the Oral Mucositis Assessment Scale (OMAS) [19]. The WHO scale measures anatomical, symptomatic as well as functional components of oral mucositis over a range of values from 0 (absence of mucositis) to 5 (when oral feeding is not possible). Mucositis was deemed severe if the WHO score was ≥3. The OAG represents a comprehensive instrument that assesses both oral cavity function and its physical aspect with multiple variable scales ranging between 8 and 24, where a score higher than 16 indicates a severe oral condition. The OMAS is a scoring system assessing the anatomic extent and severity of oral mucositis, evaluating multiple regions of the oral cavity for erythema using a 3-point scale and the presence and size of ulcerations or pseudo membranes using a 4-point scale. The mean mucositis score ranges from 0 to 5.

Secondary efficacy assessments included two patient-reported outcomes, the validated Xerostomia Questionnaire (XQ) and the EORTC QLQ-C30/QLQ-H&N35 quality of life measures.

The XQ was administered concomitantly to the evaluation of mucositis; it estimates the level of dryness by rating eight items on an 11-point ordinal Likert scale from 0 to 10. The sum of all the item scores is transformed linearly to produce the final summary score ranging from 0 to 100, with higher scores representing greater levels of xerostomia [20].

Quality of life (QoL) was assessed at baseline, twice during treatment (weeks 4 and 7), and 4 weeks (week 11) and 3 months (week 19) after the end of radiotherapy. The EORTC QLQ-C30 [13,14] is a generic cancer health-related QoL (HRQOL) questionnaire consisting of 30 items that include a scale measuring the global health status/HRQOL; five functioning scales, including physical, role, emotional, cognitive and social; three symptom scales, including fatigue, nausea/vomiting, and pain; and six single-item scales, including dyspnea, insomnia, appetite loss, constipation, diarrhea and financial impact.

The QLQ-H&N35 [15,16] is a cancer disease-specific questionnaire consisting of 35 items. These are grouped into seven symptom scales, including pain, swallowing, senses problems, speech problems, trouble with social eating, trouble with social contact and less sexuality, and 11 single-item scales, including teeth, opening mouth, dry mouth, sticky saliva, coughing, feeling ill, pain killers, nutritional supplements, feeding tube, weight loss and weight gain.

The items on both measures were scaled and scored using the recommended EORTC procedures. Raw scores were transformed to a linear scale ranging from 0 to 100, with a higher score representing a higher level of functioning or a higher level of symptoms. Provided that at least half of the items in the scale were completed, the scale score was calculated using only those items for which values existed.

Safety and tolerability were assessed by patients monitoring for adverse events at baseline, weekly during treatment and 4 weeks and 3 months after the end of radiotherapy, using the National Cancer Institute Common Terminology Criteria for Adverse Events v.4 (NCI.CTCAE v4).

At mucositis onset (any grade), all patients were provided conventional drugs for mucositis, consisting of anti-inflammatory agents, antimicrobials, coating agents and analgesics. Low-level Laser Therapy (LLLT) was allowed in case of severe mucositis onset.

### 2.5. Statistical Analysis

A sample size of 60 patients in each arm was estimated to provide 80% power at a two-sided significance level of 0.05 to detect an odds ratio of 0.33 if the severe mucositis rate was 75% in the placebo group and 50% in the SAMITAL group. Since no adjustment was made for the significance of the secondary endpoints, all secondary analyses should be interpreted as exploratory.

The occurrence of at least one event scored as 3 or 4 on the WHO scale and at least one event of OAG greater than 16 at any time post-baseline until the final visit were analyzed using a logistic regression model with the treatment arm as a categorical independent variable. Results were reported as odds ratios (ORs) with their 95% CIs.

Time to the onset of severe mucositis, defined as the time from randomization to the time of the first occurrence of WHO score 3 or 4 and time to the onset of OAG greater than 16, was estimated by Kaplan–Meier method and comparisons between arms were performed using the log-rank test.

Linear mixed-effects models were fitted over time using treatment to assess differences between the two treatment arms over time regarding OMAS, XQ and HRQOL. The time effect and treatment–time interaction were fixed effects, and the intercept and slope (unstructured covariance matrix) were random effects.

Compliance and use of conventional drugs for mucositis were analyzed by generalized linear mixed models with Poisson and binomial distributions, respectively, with treatment, time and treatment–time interaction as fixed effects and random effects for the intercept and slope (unstructured covariance matrix).

Continuous variables were described using the mean, standard deviation and quartile when appropriate. Nominal and ordinal variables were described using contingency tables.

All analyses were conducted with SAS software, version 9.4.

## 3. Results

### 3.1. Population

From December 2012 to July 2016, 120 patients entered the study protocol. Four patients had rapid deterioration of medical conditions before starting treatment and were excluded from the analysis. Out of the 116 assessed patients, 85 (43 in the SAMITAL group and 44 in the placebo group) completed the randomized treatment up to 11 weeks (Figure 1).

Patient characteristics were well-balanced between groups (Table 1).

Most patients (58.6%) received more than four cycles of concomitant chemotherapy (59.3% in the SAMITAL group and 57.9% in the placebo group) and were treated by cisplatin (47.4%), carboplatin (31.9%) or other drugs (20.7%), similarly in both groups. The radiotherapy regimen was IMRT for 104 patients (89.7%). Additionally, 111 patients (95.7%) received more than 30 fractions with a median delivered dose of 66 Gy (60–66) in the SAMITAL group and 66 Gy (60–70) in the placebo group. Only five patients (three in the SAMITAL group and two in the placebo group) had a definitive radiotherapy interruption; among them, only one patient in the SAMITAL group dropped out of radiotherapy at the 17th fraction due to grade 3 mucositis.

### 3.2. Primary Outcome

WHO grade 3–4 mucositis occurred in 60 out of 116 evaluated patients (27 patients (45.8%) in the SAMITAL group and 33 patients (57.9%) in the placebo group (OR = 0.6, 95% CI: 0.3–1.3; *p* = 0.1922)). Thirty-one patients had at least one occurrence of a severe oral condition measured by the OAG score during chemo-radiotherapy treatment (13 (22.0%) in SAMITAL and 18 (31.6%) in the placebo group (OR = 0.6, 95% CI: 0.3–1.4; *p* = 0.2475)).

Interestingly, only a few grade 4 mucositis cases were observed in patients taking SAMITAL than in the placebo group (3/27 in the SAMITAL arm vs. 14/33 in the placebo arm) (Table 2).

An explorative analysis by subgroups highlighted a statistically significant lower incidence of severe mucositis in patients randomized to receive SAMITAL (38.9%) compared with placebo (68.6%) in the subgroup of concomitant chemotherapy (OR = 0.29; 95% CI: 0.11–0.78, *p* = 0.0136).

### 3.3. Secondary Outcomes

Time to onset of WHO severe mucositis was not significantly different between the two arms (log-rank test, *p* = 0.2429; Figure 2A), but a positive trend was observed after SAMITAL treatment. At the end of radiotherapy delivery, 70.1% (95% CI: 56.3–80.2) of patients were severe mucositis free in the SAMITAL group and 58.4% (95% CI: 44.3–70.1) in the placebo group. No statistically significant difference between arms was detected on the OAG scale (log-rank test, *p* = 0.3216; Figure 2A).

Both the OMAS and the XQ significantly deteriorated over time, but the longitudinal analysis did not show a statistically significant difference between the two treatment arms. The maximum difference was registered in the XQ at week 7, when radiotherapy delivery ended (estimated mean difference: −10.7; 95% CI: −18.4 to −3.1, *p* = 0.0063), with a statistically significant lower level of xerostomia in the SAMITAL group (Figure 2B).

The overall test for differences in HRQOL scores between the two treatment arms resulting from the longitudinal mixed-effects analysis was not statistically significant for any scale. Differences assessed at each time point showed a higher QoL status (estimated mean difference: 9.3; 95% CI: 0.7–17.9; *p* = 0.0333), emotional functioning (8.0; 95% CI: 0.3–15.7; *p* = 0.0427), a lower cough (−18.5; 95% CI: −28.1 to −8.9; *p* = 0.0002), feeling ill (−8.0; 95% CI: −14.4 to −1.5; *p* = 0.0162) and swallowing score (−14.6; 95% CI: −24.1 to −5.1, *p* = 0.0026) after SAMITAL treatment at week 7. Moreover, patients receiving SAMITAL reported lower use of a feeding tube at week 4 of radiotherapy (*p* = 0.0175) and at week 11 (*p* = 0.0184). Results over time are reported in Figure 3.

### 3.4. Standard Treatments for Mucositis and Body Weight Changes

Use of non-steroidal anti-inflammatory drugs (NSAID) or opioids to relieve oral mucositis pain during the study period was not different between the two arms (overall *p* = 0.2135), while the need for artificial nutrition was significantly lower in the SAMITAL than in the placebo group (overall *p* = 0.0303, Figure 4A). Over time, there was a significant increase in NSAID/opioids administration in both arms (*p* < 0.0001), whereas artificial nutrition significantly increased from baseline only for patients treated with the placebo (*p* = 0.0004, Figure 4A).

The overall test for differences and differences assessed at each time point in body weight between the two treatment arms (Figure 4B) were not statistically significant (estimated overall mean difference: −0.8 kg; 95% CI: −6.1 to 4.5, *p* = 0.7665). Over time, body weight significantly decreased compared with baseline weight, starting from week 3 for the SAMITAL group (−1.4 kg, 95% CI: −2.6 to −0.3, *p* = 0.0118) and from week 1 for patients who received placebo (−1.7 kg, 95% CI: −2.8 to −0.5, *p* = 0.004).

### 3.5. Compliance

The overall compliance with SAMITAL/placebo (Figure 5), defined as the number of sachets returned, was lower in the placebo group (estimated mean value: 4.5 sachets; 95% CI: 3.2–6.2) than in the SAMITAL group (2.8; 95% CI: 2.0–4.0), although not statistically different (*p* = 0.0562). The greatest difference was observed during the third week of treatment when patients in the placebo group returned 4.4 sachets (95% CI: 3.0–6.4) versus 2.2 (95% CI: 1.4–3.4) in the SAMITAL group (*p* = 0.0187).

## 4. Discussion

Mucositis is a well-known modern iatrogenic illness [21]. Virtually all patients treated with radiotherapy for HNSCC develop oral mucositis; its incidence and intensity change according to cancer subsite, radiotherapy fields, dose, fractionation and association with chemotherapy [22]. Subjective factors are also involved in mucositis onset: poor oral hygiene, periodontal disease, chronic alcohol consumption, cigarette smoking, xerostomia and BMI < 18.5, as well as comorbidities such as diabetes mellitus [23]. Barasch [24] reported that age, sex and therapeutic regimen are additional risk factors for mucositis onset, but nowadays, the relative contribution of causes for mucositis and a comprehensive understanding of its pathogenesis is still under investigation. Antimicrobials, anesthetics, analgesics and LLLT (only for severe mucositis) are usually employed but have not solved oral mucositis-related problems. Oral care, anti-inflammatory and natural agents are often recommended to improve symptoms [25].

It is difficult to precisely assess the real incidence of oral mucositis in HNSCC (chemo)-radiotherapy-treated patients. The reported incidence of severe oral mucositis has changed over time, especially over the past twenty years, depending on the employment of different radiation techniques and scoring systems.

A review of 33 studies published from 1996–1999 reported an incidence of severe mucositis ranging from 34% for patients receiving conventional radiotherapy for HNSCC to 56% for patients receiving altered fractionation radiotherapy [26]. The authors reported that the WHO scale was the most frequently used (less than half of the studies), followed by the Radiation Therapy Oncology Group instrument (RTOG). The incidence of mucositis was increased by the introduction of concurrent chemotherapy, as well as by altered fractionation schedules [26]. Considering 245 patients with stage III and IV HNSCC treated with primary (chemo)-radiotherapy (135 patients treated with C3DRT, 110 patients with IMRT), Lambrecht et al. reported that patients treated with IMRT developed significantly less severe acute oral mucositis (CTCAE version 3.0) than those who underwent C3DRT (32 vs. 44%) [27]. On the contrary, in a Phase 2 study of definitive chemo-radiotherapy using only an induction protocol, Milano et al. did not find a significant difference in acute severe mucositis between patients treated with IMRT (85%) or C3DRT (79.2%) [28].

In our study, the IMRT radiation technique was employed in 89.8% of patients treated with SAMITAL and in 89.5% of those treated with placebo. The frequency of severe mucositis was lower than expected in both arms (45.8% for SAMITAL vs. 57.9% for placebo), thus suggesting that IMRT may cause less frequent acute severe mucositis (grade ≥ 3 WHO and OAG) than C3DRT, as also reported by Dijkema et al. [29]. Furthermore, the difference in incidences of severe oral mucositis between the two arms was 12.1%, which was lower than the expected percentage of approximately 25%.

Previous studies that reported relevant reductions in mucositis grade with decreased pain and improvement in QoL in adult patients treated with SAMITAL for chemo-radiotherapy-induced oral mucositis [10,12] are available. In our study, only a few grade-4 mucositis cases were observed after SAMITAL administration, with respect to placebo treatment. The limited difference in severe oral mucositis incidence (sum of grades 3–4) between the two arms may be explained by multiple factors. First, poor drug conservation, preparation by patients and challenges in the correct mode of administration might have modified the drug’s efficacy. SAMITAL or its related matching placebo have to be taken orally and requires careful attention in the preparation and administration, while the self-awareness of patients tends to worsen due to the effects of anti-cancer medical treatments. For SAMITAL (and the related matching placebo), the need to respect the posology, the dosage, and the drug’s preparation according to label instructions can also be demanding for patients. SAMITAL (and the related matching placebo) is required to be stored in a refrigerator at +5 degrees (range +2–8), reconstituted with 20 mL water, stirred, thickened, taken in small amounts at a time and kept in the mouth for at least 1 min. The patient has to spit out or swallow the liquid within 30 min without rinsing or drinking for 10 min after the procedure. In our study, this procedure had to be performed four times a day during the whole radiotherapy period and 28 days after. Moreover, arising odynophagia and dysphagia caused by chemo-radiotherapy treatments could also represent an obstacle in SAMITAL correct administration. Approximately 64% of HNSCC patients treated by chemo-radiotherapy show dysphagia during treatment [30], therefore increasing the difficulty in spitting out or swallowing an oral product. Second, mucositis grade might have been underestimated during clinical observation: clinician subjectivity cannot be excluded even in double-blinded studies. Third, a bias may be present between the two arms in terms of smoking and alcohol use during the study since HNSCC patients often do not stop such habits during (chemo)-radiotherapy and these carcinogenic agents are associated with mucositis.

Moreover, patients aware of causative factors of disease tend to underestimate habits in clinical reporting

Furthermore, the statistically significant lower incidence of severe mucositis in patients randomized to receive SAMITAL in the subgroup of concomitant chemotherapy deserves further study.

SAMITAL was well-tolerated, and the taste was considered acceptable even at the beginning of the treatment when patients were not affected by taste modifications. SAMITAL was also safe, and no local or systemic pharmacological, allergic, toxic or synergistic/antagonistic side effects were recorded. Episodes of nausea and vomiting that occurred in some cases were considered related to chemotherapy.

Moreover, SAMITAL provided a better QoL compared to placebo, as shown by the results of XQ, EORTC QLQ-C30 and QLQ-C H&N35. Regarding xerostomia, patients sensed the therapeutic effect of SAMITAL, especially on week 7 when they had received the whole dose of radiotherapy. This effect also continued during the following 4 weeks (even if the statistical results at week 11 were not significant). The improvement in xerostomia is important as salivary gland hypofunction, usually accompanied by a persistent feeling of dry mouth, might seriously increase the risk of developing oral infections and tooth decay, oral mucosal discomfort and pain, hampered oral functioning and a worsened nutritional state [31]. Xerostomia, as well as dysphagia, can also persist as a delayed effect of radiotherapy, worsening the QoL of surviving patients for years after the end of the treatment [31].

Specific items of EORTC showed that SAMITAL was effective when the patients were more hard-pressed. More in detail, global health status, emotional functioning, as well as dysphagia-related symptoms were better at the end of radiotherapy (week 7) when the whole dose of RT was delivered (week 7) with SAMITAL when compared with placebo. Cough, feeling ill, dysphagia, nutritional supplement need and necessity for a feeding tube were also less frequent in SAMITAL treatment than in the placebo group. Considering that severe mucositis becomes apparent starting from week 4 [32], it is interesting to note that on week 5, the need for artificial nutrition started to be significantly higher in the placebo arm compared with SAMITAL (Figure 4B), suggesting a positive effect of SAMITAL exactly when mucositis symptoms begin, as reported by patients.

Despite the possibility of interrupting radiation treatment due to mucositis, only one patient in our study (in the SAMITAL group) definitively interrupted radiotherapy due to toxicity. The low incidence of definitive interruption can be explained by the use of IMRT, which might have decreased the severity of oral mucositis, and by the possibility of performing a weekly otolaryngological examination, which enabled patients to be promptly admitted to day hospital care at the onset of severe mucositis onset, and to be hospitalized in case of significant worsening.

Our clinical study has some strong points together with some limitations.

The main strengths of our investigation were: the consecutive and numerous cases, the strong exclusion and inclusion criteria, the rigorous design of the trial (which is a prospective, randomized, double-blind and placebo-controlled investigation), the histological homogeneity of cases, the homogeneous staging (only III–IV M0 HNSCC) and the fact that radiation treatment was performed in the same center by the same staff. Moreover, weekly endoscopic otolaryngological evaluation for mucositis, a multidisciplinary approach and the use of a validated questionnaire that was provided to patients to assess xerostomia and their quality of life enabled a careful and continuous evaluation of the patients’ clinical conditions.

Some of the limitations of this study were: the inclusion of different sub-sites of HNSCC (oral, oropharyngeal, larynx and hypopharyngeal), which might have altered the concentration and effects of SAMITAL, especially in different areas depending on sub-sites, and also due to the possibility of the patient’s choice to split or swallow SAMITAL/placebo; finally, the fact that chemotherapy protocol was not performed by the same team, and a possible subjectivity bias in the clinical evaluation of mucositis. Moreover, before the beginning of the study, the estimated incidence of mucositis was higher than measured in our study, based on our knowledge of previous reports, thus reducing the possibility of seeing a significant effect of SAMITAL in the whole sample compared with the placebo.

SAMITAL effects should be further investigated on larger series of patients, and HNSCC patients (staged according to AJCC 8th edition) should be selected by the site of disease (for example, oral and oropharynx), type of radiotherapy (or IMRT/C3DRT) and chemotherapy. We also suggest that chemo-radiotherapy treatment should be planned and performed for all patients by the same radiotherapist and oncologist staff and that statistical design should consider a lower incidence of oral mucositis in IMRT cases compared with previous C3DRT cases. Additionally, SAMITAL should be prepared and delivered by a caregiver, and an additional dose should be taken before the patient falls into nocturnal sleep. Many patients complain of nocturnal awakening from a dry mouth and also after parotid-sparing radiotherapy for HNSCC, which can be caused by submandibular gland dysfunction [33]. Furthermore, it would be easier for patients not to swallow SAMITAL but use it only for mouth rinses to have the best effect on occurring oral mucosa as the optimal possibility for oral and oropharyngeal cancer patients

Moreover, to confirm the effect of SAMITAL on dysphagia, it could be interesting to study its effect on patients suffering from esophageal squamous cell carcinoma submitted to a chemo-radiotherapy protocol. In this population, it would also be important to reduce the number of administrations required and improve the pharmaceutical formulation with a lower density to make it easier to swallow.

## 5. Conclusions

Radiotherapy-induced mucositis is still worsening patients’ QoL; therefore, mucositis treatment must be improved. SAMITAL, a botanical drug containing three highly standardized extracts, might be beneficial for patients suffering from chemo-radiotherapy-induced oral mucositis. The results of our study did not show a significant superiority of SAMITAL compared with the placebo in reducing the incidence of severe mucositis; however, in the subgroup analysis of patients treated with SAMITAL and concomitant chemotherapy, a statistically significant improvement of severe mucositis was noted. Moreover, all SAMITAL-treated patients reported an improvement in QoL compared with placebo, as measured by XQ, EORT-QLQC30 and EORTC-HN35, thus suggesting a positive effect of SAMITAL in preventing or treating early and late symptoms of oral mucositis.

Further studies on SAMITAL are mandatory to adjust the dosage and formulation suggested and to focus on the target population.

Investigators must continue to seek more effective therapy for this patient population.

## Figures and Tables

**Figure 1 cancers-14-06192-f001:**
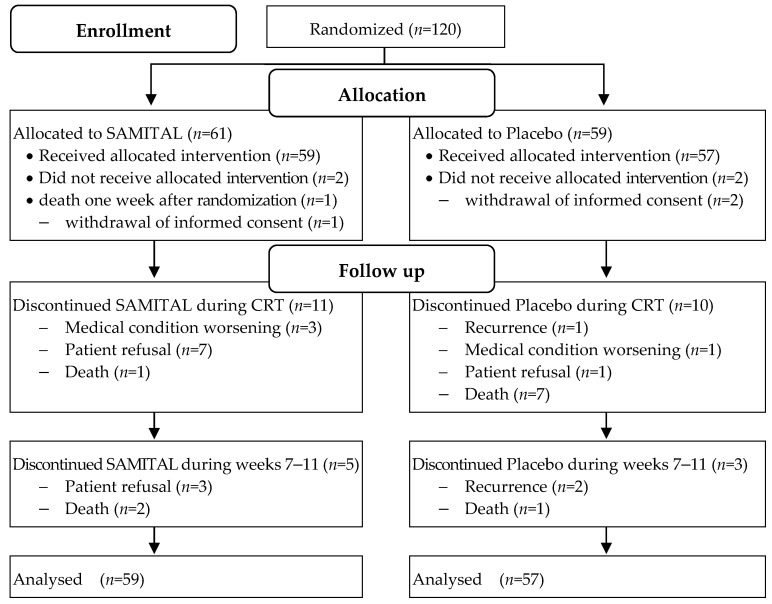
Trial flow chart.

**Figure 2 cancers-14-06192-f002:**
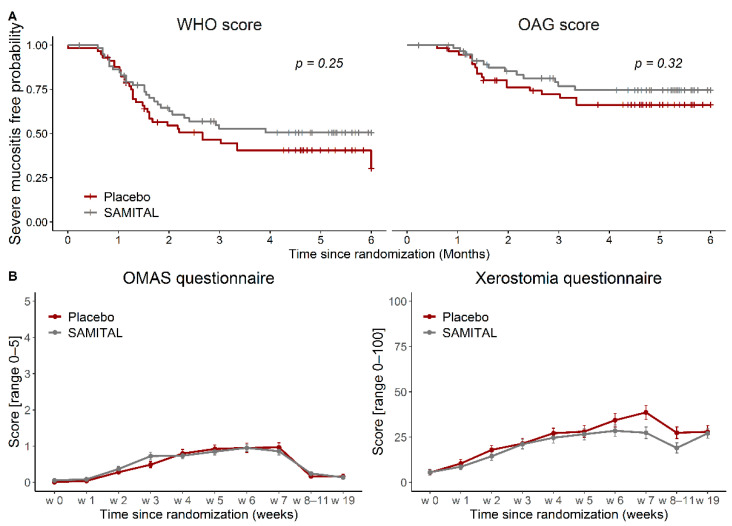
(**A**) Severe mucositis-free probability as time from randomization to the occurrence of a grade 3/4 mucositis, as measured by the WHO score, or to the occurrence of a severe oral condition, as measured by an OAG score > 17. (**B**) Scores over time for the OMAS and the Xerostomia Questionnaire. Data are presented as estimated mean scores (repeated-measure models) at every time point, with their 95% confidence interval. A higher score indicates a worse oral condition. WHO, World Health Organization; OAG, Oral Assessment Guide. OMAS, Oral Mucositis Assessment Scale.

**Figure 3 cancers-14-06192-f003:**
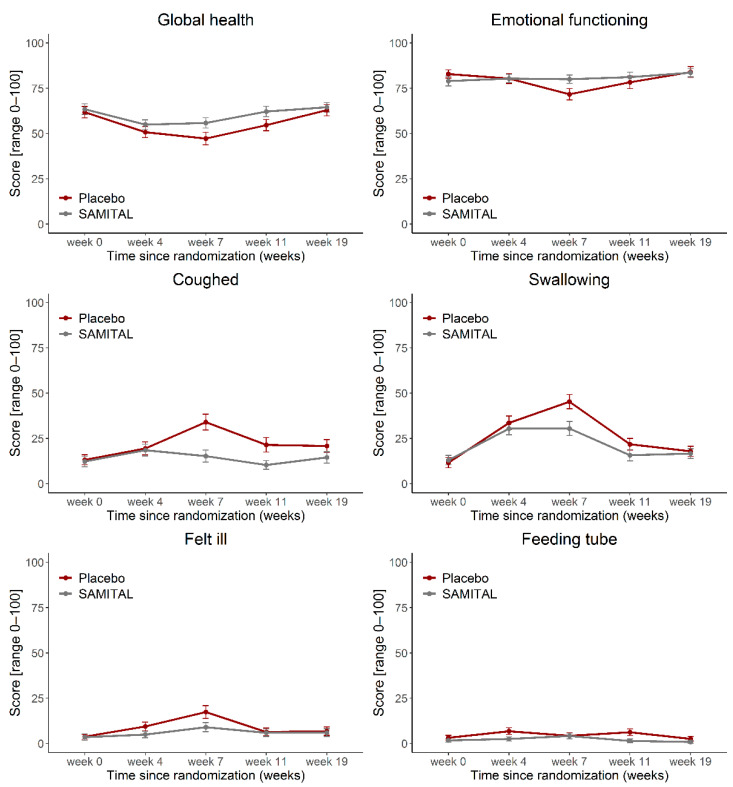
Health-related Quality of Life (HRQoL) scores over time for selected EORTC QLQ-C30 functional scales and EORTC QLQ-H&N35 symptom items/scales. Data are presented as estimated mean HRQoL scores (repeated-measure mixed-effect models) at every time point, together with their 95% confidence intervals. A higher score represents a higher level of functioning or symptoms. EORTC, European Organization for Research and Treatment of Cancer; QLQ-C30, Quality of Life Questionnaire Core 30; QLQ-H&N35, Quality of Life Questionnaire head and neck cancer module.

**Figure 4 cancers-14-06192-f004:**
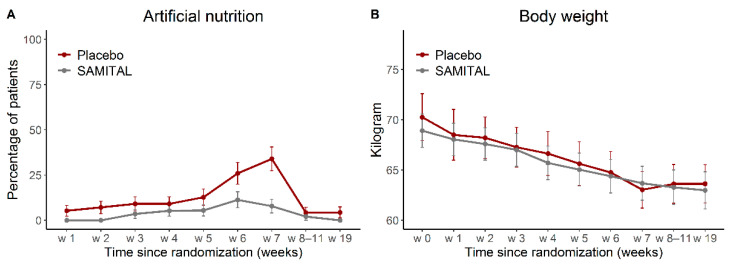
(**A**) Use of artificial nutrition and (**B**) body weight over time. Data are presented as estimated values at every time point, together with their 95% confidence interval.

**Figure 5 cancers-14-06192-f005:**
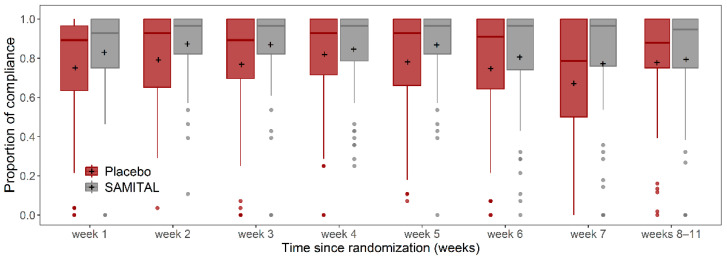
Box plot of overall compliance, defined as the number of sachets consumed out of those planned during radiotherapy (weeks 1 to 7) and 4 weeks after the end of radiotherapy (week 11). The heavy horizontal line represents the median; the box represents the interquartile range; the whiskers represent 95% confidence intervals; the circles represent outliers.

**Table 1 cancers-14-06192-t001:** Clinical and cancer treatment characteristics by the randomized arm.

Clinical Characteristics		Treatment		
		SAMITAL (*n* = 59)	Placebo (*n* = 57)	*p*-Value
Sex, *n* (%)	F	14 (23.7)	17 (29.8)	0.4583
	M	45 (76.3)	40 (70.2)	
Age, years	Median (Q1–Q3)	65.1 (58.8–72.2)	66.0 (60.9–73.3)	0.3535
Weight loss, kilograms (in the last 3 months)	Median (Q1–Q3)	0 (0.0–7.0)	0 (0.0–5.0)	0.8231
Karnofsky PS, *n* (%)	70	0 (0.0)	1 (1.8)	0.7905
	80	21 (35.6)	20 (35.1)	
	90	20 (33.9)	19 (33.3)	
	100	18 (30.5)	17 (29.8)	
Cigarette smoke, *n* (%)	No	7 (11.9)	12 (21.1)	0.1030
	Yes	21 (35.6)	11 (19.3)	
	Former smoker	31 (52.5)	34 (59.6)	
Alcohol, *n* (%)	No	22 (37.3)	21 (36.8)	0.3228
	Yes	24 (40.7)	29 (50.9)	
	Former alcoholic	13 (22.0)	7 (12.3)	
Previous illness	Cardiovascular	14 (23.73)	16 (28.07)	0.6752
	Hypertension	25 (42.37)	24 (42.11)	1.0000
	Diabetes	7 (11.86)	6 (10.53)	1.0000
	Respiratory	0 (0.00)	4 (7.02)	0.1184
Site of disease, *n* (%)	Oral cavity	16 (27.1)	9 (15.8)	0.5193
	Oropharynx	26 (44.1)	30 (52.6)	
	Hypopharynx	8 (13.6)	8 (14.0)	
	Larynx	9 (15.3)	10 (17.5)	
Stage of disease, *n* (%)	III	8 (13.6)	8 (14.0)	0.9408
	IV	51 (86.4)	49 (86.0)	
Previous surgery for the current disease	Yes	22 (37.3)	20 (35.1)	0.8053
Chemotherapy, *n* (%)	Concomitant	36 (61.0)	35 (61.4)	0.9659
	Induction + concomitant	23 (39.0)	22 (38.6)	
Chemotherapy type, *n* (%)	Cisplatin	30 (50.8)	25 (43.9)	0.5319
	Carboplatin	16 (27.1)	21 (36.8)	
	Other drugs	13 (22.0)	11 (19.3)	
Radiotherapy type, *n* (%)	C3DRT	6 (10.2)	6 (10.5)	0.9497
	IMRT	53 (89.8)	51 (89.5)	
Radiotherapy	No	56 (94.9)	55 (96.5)	0.6761
interruption, *n* (%)	Definitive	3 (5.1)	2 (3.5)	
Reason for radiotherapy	Toxicity	1		
interruption, *n* (%)	Other	2	2	
Dose delivered, Gy	Median (Q1–Q3)	66 (60–66)	66 (60–70)	0.4193

**Table 2 cancers-14-06192-t002:** Incidence of WHO mucositis by the randomized arm. WHO, World Health Organization.

		Treatment	
		SAMITAL	Placebo
WHO mucositis grade, *n* (%)	0	2 (3.4)	1 (1.7)
	1	4 (6.8)	6 (10.5)
	2	26 (44.1)	17 (29.8)
	3	24 (40.7)	19 (33.3)
	4	3 (5.1)	14 (24.6)
WHO severe mucositis grade, *n* (%)	3–4	27 (45.8)	33 (57.9)

## Data Availability

Research data are stored in an institutional repository and will be shared upon reasonable request to the corresponding author.

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
