# Peer review of "Role of SAMITAL in the Prevention and Treatment of Chemo-Radiotherapy-Induced Oral Mucositis in Head and Neck Carcinoma: A Phase 2, Randomized, Double-Blind, Placebo-Controlled Clinical Trial (ROSAM)"

_cancers, 2022, doi:10.3390/cancers14246192_

Round 1

Reviewer 1 Report

i feel that this was a negative study .  None the less you have done an elegant clinical trial which is a model for others to follow .  

i would ask that you conclude that   " investigators must continue to seek more effective therapy for this patient population "

Author Response

Thank you for the suggestion, we have now included the suggested statement in the Conclusion section.

Reviewer 2 Report

Methods

1)    No mentions are present about the flavor of the drug compared to the placebo. It should be specified given the fact that it could be an important factor regarding the compliance of the patients.

2)    The composition of the drug is well described, but no mentions are present about the posology and the methods of administration. For instance, only in the results we become aware of the fact that each patient can freely chose either to swallow or spit the drugs after rinsing it in the mouth.

3)    Patients with previous surgery were also excluded?

Results

1)    In figure 1 we can read that 3 patients in SAMITAL group and 1 patient in Placebo group were advised by the investigator to discontinue the treatment (or placebo). The motivation behind those decisions should be specified. 

2)    The authors state that “Most patients... were treated by cisplatin (47.5%), carboplatin (32.2%) or other drugs (20.3%), similarly in both groups”, but the differences in treatment between the two arms are not specified.

3)    As stated in the discussion, clinical comorbidities as diabetes or BMI<18.5 have an important role in the incidence of mucositis after chemo-radiotherapy. A part weight loss and PS, no comorbidities or clinical condition are specified and therefore no differences between the two arms are deductible.

4)    The authors declare that “At mucositis onset (any grade) all patients were provided conventional drugs for mucositis” but then no data are shown about the need of those treatment in the two arms. Differences in the use of other treatment between the two arms would strongly impacts the results.

5)    The overall compliance is distinctly different between the two arms, as also stated by the authors. This could be the major limitation of the study considering that most of the benefits of SAMITAL were seen at 4 weeks, just the week after a significant difference in compliance was found (third week).

Discussion

1)    Posology and methods of administration of SAMITAL and Placebo should be moved to the Method section. 

2)    We can read “it is interesting to note that on week 5, the need for standard mucositis treatment was significantly lower in SAMITAL arm compared to placebo (89.1% vs 70.9%, p=0.0171, data not shown)”. Those data, argued more in depth, should be moved to the Result section.

References

1)    Many references are missing.

Author Response

Comments and Suggestions for Authors

Methods

1)    No mentions are present about the flavor of the drug compared to the placebo. It should be specified given the fact that it could be an important factor regarding the compliance of the patients.

Flavor matching between Verum and Placebo was obtained by using the same sweetening agents in the two compositions, described in Material and Methods. A statement was added in the relevant section regarding masking.

2)    The composition of the drug is well described, but no mentions are present about the posology and the methods of administration. For instance, only in the results we become aware of the fact that each patient can freely chose either to swallow or spit the drugs after rinsing it in the mouth.

Thank you for the suggestion, we have now included administration and posology in Material and Methods, accordingly.

3)    Patients with previous surgery were also excluded?

Indeed, patients with previous surgery were eligible for the study, as well. We have now included information about the number of patients who underwent previous surgery and a comparison test in Table 1.

Results

1)    In figure 1 we can read that 3 patients in SAMITAL group and 1 patient in Placebo group were advised by the investigator to discontinue the treatment (or placebo). The motivation behind those decisions should be specified. 

Patients were advised by the investigator to discontinue the treatment for deterioration of medical conditions. These four patients also discontinued the radiotherapy treatment. This was specified in figure 1, as requested

2)    The authors state that “Most patients... were treated by cisplatin (47.5%), carboplatin (32.2%) or other drugs (20.3%), similarly in both groups”, but the differences in treatment between the two arms are not specified.

Thank you for the suggestion, we have now included chemotherapy type and the comparison test between the two arms in Table 1.

3)    As stated in the discussion, clinical comorbidities as diabetes or BMI<18.5 have an important role in the incidence of mucositis after chemo-radiotherapy. A part weight loss and PS, no comorbidities or clinical condition are specified and therefore no differences between the two arms are deductible.

Thank you for the suggestion, we have now included comorbidities and comparison test between the two arms in Table 1.

4)    The authors declare that “At mucositis onset (any grade) all patients were provided conventional drugs for mucositis” but then no data are shown about the need of those treatment in the two arms. Differences in the use of other treatment between the two arms would strongly impacts the results.

We would like to thank the reviewer for this important comment. We added a paragraph detailing the use of other standard treatments for the mucositis during the study period. On the base of this new analysis we modified the discussion accordingly.

5)    The overall compliance is distinctly different between the two arms, as also stated by the authors. This could be the major limitation of the study considering that most of the benefits of SAMITAL were seen at 4 weeks, just the week after a significant difference in compliance was found (third week).

Overall, compliance was higher in the SAMITAL arm, however, as reported in the results section, this was not statistically different respect to the placebo arm. Only at week 3 this difference was statistically significant. We don’t think this could be considered a limitation of the study, considering that the study was a double blind placebo controlled randomized trial. 

Discussion

  • Posology and methods of administration of SAMITAL and Placebo should be moved to the Method section. 

As suggested by the reviewer, we described posology and methods of administration more in depth in the method section

2)    We can read “it is interesting to note that on week 5, the need for standard mucositis treatment was significantly lower in SAMITAL arm compared to placebo (89.1% vs 70.9%, p=0.0171, data not shown)”. Those data, argued more in depth, should be moved to the Result section.

As suggested by the reviewer, we analysed the use of standard mucositis treatment and reported the results in a new paragraph added in the results section.

References

1)    Many references are missing.

We really apologize for this. We did not realize that many references had been deleted. We have now included them correctly.

Reviewer 3 Report

Your article about “Role of SAMITAL in prevention and treatment of chemo-radiotherapy-induced oral mucositis in head and neck carcinoma: A Phase 2, randomized, double blind, placebo-controlled clinical trial (ROSAM)” is very interesting. 

There are some questions for your article : 

  1. Moderate and severe mucositis pain can relief with good oral hygiene. In your study, patients were advised about oral hygiene. Please describe the specific practice about keep oral hygiene.

  2. In your study, 7 patients in SAMITAL group and 1 patient in placebo group discontinued medication. Please describe the reason why these patients discontinued medication.

  3. As stated above, please described why the investigator decided to discontinue medication.

  4. The severity of oral mucositis will be affected by comorbidity, such as DM, HTN, and hepatitis. Please provide the detail of comorbidity in table 1.

  5. The SAMITAL required to be stored in the refrigerator. How do you confirm that all these patients cooperated with the storage method and steps to use SAMITAL?

  6. Oral mucositis pain is an anticipated complication of cancer therapy. Severe oral mucositis may affect oral intake and body weight. Please provide the body weight record before, during, and after the study. And you should assess body weight change between to groups and characterize statistical outcomes.

  7. Basically, we use NSAID or opioids to relieve oral mucositis pain. With the assistance of analgesic drugs, patients may improve their quality of life, and swallowing function and remove the feeding tube. Please describe the medication about the analgesic drug (what kind of drug and daily dosage) and characterize the statistical outcome in two groups.

  8. For OMAS, XQ and HRQOL, the authors used a linear mixed effects model, however, the linear mixed effects model requires a normality assumption for the dependent variable. Have you checked that OMAS, XQ and HRQOL are normal?

  9. Can the authors explain why the compliance proportions are fitted using a generalized linear mixed model with a beta distribution, but not other distributions, such as binomial or Poisson?

  10. Please provide p-values in Table 1, even if they are well-balanced.

  11. In the Statistical Analysis section, please add the statistical method used to compare the survival distribution between SAMITAL and placebo.

Author Response

There are some questions for your article : 

  1. Moderate and severe mucositis pain can relief with good oral hygiene. In your study, patients were advised about oral hygiene. Please describe the specific practice about keep oral hygiene.

As suggested by the reviewer, we added a statement in the method section regarding oral hygiene.

  1. In your study, 7 patients in SAMITAL group and 1 patient in placebo group discontinued medication. Please describe the reason why these patients discontinued medication.

Unfortunately, patients did not give further details about the reason to discontinue medication.

  1. As stated above, please described why the investigator decided to discontinue medication.

Patients were advised by the investigator to discontinue the treatment for deterioration of medical conditions. These patients also discontinued the radiotherapy treatment. This was specified in figure 1.

  1. The severity of oral mucositis will be affected by comorbidity, such as DM, HTN, and hepatitis. Please provide the detail of comorbidity in table 1.

Thank you for the suggestion, we have now included comorbidities and comparison test between the two arms in Table 1.

  1. The SAMITAL required to be stored in the refrigerator. How do you confirm that all these patients cooperated with the storage method and steps to use SAMITAL?

Patients were instructed on treatment preparation, administration and storage, and were asked to confirm the procedures used at home during each visit. To note, the indication to store the product in refrigerated conditions was a prudential decision pending the completion of stability studies on SAMITAL sachets. By evaluating the positive results of these stability studies, once completed, it would have been possible to consider also a storage of the product at room temperature, especially taking into account the duration of the clinical study.

  1. Oral mucositis pain is an anticipated complication of cancer therapy. Severe oral mucositis may affect oral intake and body weight. Please provide the body weight record before, during, and after the study. And you should assess body weight change between to groups and characterize statistical outcomes.

Thanks, this is a good point. We have reported analysis regarding body weight changes during the treatment period and between the two arms.

  1. Basically, we use NSAID or opioids to relieve oral mucositis pain. With the assistance of analgesic drugs, patients may improve their quality of life, and swallowing function and remove the feeding tube. Please describe the medication about the analgesic drug (what kind of drug and daily dosage) and characterize the statistical outcome in two groups.

As suggested, we added a paragraph about use of standard mucositis treatment.

  1. For OMAS, XQ and HRQOL, the authors used a linear mixed effects model, however, the linear mixed effects model requires a normality assumption for the dependent variable. Have you checked that OMAS, XQ and HRQOL are normal?

We graphically checked the distribution of each response variable, conditional on the random effects, and the studentized residuals appeared to be normally distributed.

  1. Can the authors explain why the compliance proportions are fitted using a generalized linear mixed model with a beta distribution, but not other distributions, such as binomial or Poisson?

We used a Beta distribution since we defined the compliance as the number of sachets consumed out of those planned. Non we changed the outcome and used the number of sachets returned with the Poisson distribution

  1. Please provide p-values in Table 1, even if they are well-balanced.

We have now included p-values to test differences between the two arms in Table 1.

  1. In the Statistical Analysis section, please add the statistical method used to compare the survival distribution between SAMITAL and placebo.

Thank you for the suggestion, we have now added the method used.

Round 2

Reviewer 2 Report

The authors adequately addressed my comments

Author Response

We would like to thanks again the reviewer for the valuable comments. 

Reviewer 3 Report

We appreciate the authors’ effort for replying to our concerns, especially for the detailed figure for changing of body weight during treatment.

1.      Your study seems to be the exact same study as NCT01941992 in ClinicalTrials.gov. If this is the result of NCT01941992, please declare that in your article.

2.      In table 1, you mentioned “previous operation.” Do you mean surgical treatment for the current disease, or past surgical history? Please clarify.

Author Response

We appreciate the authors’ effort for replying to our concerns, especially for the detailed figure for changing of body weight during treatment.

We would like to thank again the reviewer for the valuable comments.

  1. Your study seems to be the exact same study as NCT01941992 in ClinicalTrials.gov. If this is the result of NCT01941992, please declare that in your article.
    1. We confirm that the study was registered also in the ClinicalTrials.gov database with the code NCT01941992. We added this information along with the Eudract number in the text of the article where appropriate.
  2. In table 1, you mentioned “previous operation.” Do you mean surgical treatment for the current disease, or past surgical history? Please clarify.
    1. As requested, we clarified this aspect in the Table 1. Surgery was relative to the Head and Neck cancer.